# Mn_3_O_4_ Nanocrystal-Induced Eryptosis Features Ca^2+^ Overload, ROS and RNS Accumulation, Calpain Activation, Recruitment of Caspases, and Changes in the Lipid Order of Cell Membranes

**DOI:** 10.3390/ijms26073284

**Published:** 2025-04-01

**Authors:** Yuriy Kot, Volodymyr Prokopiuk, Vladimir Klochkov, Liliya Tryfonyuk, Pavel Maksimchuk, Andrey Aslanov, Kateryna Kot, Oleg Avrunin, Lesya Demchenko, Saulesh Kurmangaliyeva, Anatolii Onishchenko, Svetlana Yefimova, Ondrej Havranek, Anton Tkachenko

**Affiliations:** 1Department of Biochemistry, V.N. Karazin Kharkiv National, 4 Svobody sq, 61022 Kharkiv, Ukraine; kot.juriy@gmail.com (Y.K.); kate.v.kot@gmail.com (K.K.); 2Department of Cryobiochemistry, Institute for Problems of Cryobiology and Cryomedicine of the National Academy of Sciences of Ukraine, 23 Pereyaslavskaya st, 61015 Kharkiv, Ukraine; v.yu.prokopiuk@gmail.com (V.P.); ai.onishchenko@knmu.edu.ua (A.O.); 3Research Institute of Experimental and Clinical Medicine, Kharkiv National Medical University, 4 Nauky ave, 61022 Kharkiv, Ukraine; 4Department of Nanostructured Materials, Institute for Scintillation Materials of the National Academy of Sciences of Ukraine, 60 Nauky ave, 61072 Kharkiv, Ukraine; klochkov@isma.kharkov.ua (V.K.); pavel.maksimchuk@gmail.com (P.M.); aslanov@isma.kharkov.ua (A.A.); ephimova@isma.kharkov.ua (S.Y.); 5Institute of Health, National University of Water and Environmental Engineering, 11 Soborna st, 33028 Rivne, Ukraine; liliya_tryfonyuk@yahoo.pl; 6Department of Biomedical Engineering, Kharkiv National University of Radio Electronics, 14 Nauky ave, 61116 Kharkiv, Ukraine; oleh.avrunin@nure.ua; 7Department of Chemistry, Stockholm University, Svante Arrhenius väg 16C, SE-106 91 Stockholm, Sweden; lesyademch@gmail.com; 8Ye.O.Paton Institute of Materials Science and Welding, National Technical University of Ukraine “Igor Sikorsky Kyiv Polytechnic Institute”, 37 Beresteiskyi ave, 03056 Kyiv, Ukraine; 9Department of Microbiology, Virology and Immunology, West Kazakhstan Marat Ospanov Medical University, 68 Maresyev st, Aktobe 030012, Kazakhstan; saule_cc@mail.ru; 10BIOCEV, First Faculty of Medicine, Charles University, Průmyslová 595, 25250 Vestec, Czech Republic; ondrej.havranek@lf1.cuni.cz; 11First Department of Internal Medicine-Hematology, First Faculty of Medicine, Charles University, 12808 Prague, Czech Republic; 12Universal Scientific Education and Research Network (USERN), 61022 Kharkiv, Ukraine

**Keywords:** calcium signaling, cytotoxicity, eryptosis, nanoparticles, oxidative stress, regulated cell death

## Abstract

Accumulating evidence suggests that manganese oxide nanoparticles (NPs) show multiple enzyme-mimicking antioxidant activities, which supports their potential in redox-targeting therapeutic strategies for diseases with impaired redox signaling. However, the systemic administration of any NP requires thorough hemocompatibility testing. In this study, we assessed the hemocompatibility of synthesized Mn_3_O_4_ NPs, identifying their ability to induce spontaneous hemolysis and eryptosis or impair osmotic fragility. Concentrations of up to 20 mg/L were found to be safe for erythrocytes. Eryptosis assays were shown to be more sensitive than hemolysis and osmotic fragility as markers of hemocompatibility for Mn_3_O_4_ NP testing. Flow cytometry- and confocal microscopy-based studies revealed that eryptosis induced by Mn_3_O_4_ NPs was accompanied by Ca^2+^ overload, altered redox homeostasis verified by enhanced intracellular reactive oxygen species (ROS) and reactive nitrogen species (RNS), and a decrease in the lipid order of cell membranes. Furthermore, Mn_3_O_4_ NP-induced eryptosis was calpain- and caspase-dependent.

## 1. Introduction

Hemocompatibility testing is crucially important to assess the toxicity of nanomaterials administered systemically [1]. Nanomaterials are known to enhance blood clotting [2] and exert immunotoxicity [3]. Therefore, evaluation of these parameters is essential in blood compatibility testing. Additionally, erythrocyte–nanomaterial interactions are widely studied and need to be evaluated since erythrocytes are the most common blood cells in circulation and the disruption of their function has serious consequences. In particular, enhanced erythrocyte aggregation changes the rheological properties of blood, increasing its viscosity [4]. The lytic cell death of erythrocytes (hemolysis), which is frequently induced by nanomaterials [5], leads to general hemolysis-associated symptoms, including the accumulation of hemoproteins with eventual consequent hemolysis-induced kidney injury [6], and could result in the activation of the innate immunity [7]. Currently, hemolysis testing is widely accepted in nanotoxicological studies as a general marker of blood compatibility [5]. However, it has been suggested that hemolysis testing should be supplemented with the evaluation of nanomaterial-induced eryptosis for more sensitive and precise nanomaterial hemocompatibility assessments [8]. Eryptosis is a type of regulated cell death of erythrocytes, which results in cell shrinkage, membrane blebbing, phosphatidylserine externalization, and the subsequent clearance of eryptotic erythrocytes by macrophages. This type of regulated cell death is mediated primarily by Ca^2+^ overload [9,10,11,12]. Generally, eryptosis assays are more sensitive and provide accurate measurements with high reproducibility. On top of that, eryptosis relies on specific signaling pathways. This allows for more precise identification of eventual nanocytotoxicity-associated molecular pathways, which cannot be achieved by hemolysis testing [8]. Hemolysis is a type of accidental cell death occurring when red blood cells are critically mechanistically damaged. Therefore, it cannot provide a functional explanation of the source or mechanism of such damage. The higher sensitivity of eryptosis assays compared to hemolysis can be linked to the physiological functions of eryptosis. There is accumulating evidence that eryptosis develops in response to external and internal perturbations to erythrocytes, which rapidly culminates in phosphatidylserine externalization and the clearance of erythrocytes by macrophages, occurring within minutes [10]. Thus, the eryptosis of dysfunctional or damaged erythrocytes ensures the rapid clearance of these cells before cell membrane integrity is impaired and the potential detrimental release of the intraerythrocytic content can occur (hemolysis). Admittedly, eryptosis is triggered by subhemolytic lethal stimuli [8].

In this study, we evaluated the hemocompatibility of synthesized Mn_3_O_4_ nanocrystals. In general, manganese ions (Mn^2+^/Mn^3+^) play an important biogenic role, acting as co-factors of various enzymes, including one type of superoxide dismutase (Mn-SOD). Manganese ions can switch their valence reversibly, which allows for the effective conversion of superoxide ions (O_2_^−^) into hydrogen peroxide (H_2_O_2_). At the same time, excessive H_2_O_2_ is further degraded by catalase. A number of studies have shown that manganese oxide nanocrystals (MnO_2_ and Mn_3_O_4_) exert enzyme-like activity. For instance, Liu et al. reported that MnO_2_ nanocrystals exhibited peroxidase- and oxidase-like activity, stimulating the oxidation of organic molecules in the presence of oxygen dissolved in water (oxidase-like activity) or the degradation of hydrogen peroxide (peroxidase-like activity) [13]. Moreover, MnO_2_ nanocrystals also reduced the amount of hydroxyl radicals formed during the decomposition of hydrogen peroxide in an aqueous solution, which indicated that they had catalase-like activity. Notably, hydrogen peroxide decomposition and the formation of oxidation products followed Michaelis–Menten kinetics, which indicated the enzyme-like type of catalytic activity of nanocrystals [13]. Huang et al. created a nanocomposite material in the form of submicron particles based on vanadate (V_2_O_5_) and manganese oxide (MnO_2_) nanocrystals linked together by dopamine molecules. V_2_O_5_ nanocrystals demonstrated catalytic activity similar to that of glutathione peroxidase, while MnO_2_ nanocrystals elicited SOD- and catalase-like activity [14]. Thus, the resulting material exhibited activity similar to that of three enzymes at once—SOD, catalase, and glutathione peroxidase. A biological evaluation confirmed a higher antioxidant activity of the composite material in comparison to the individual MnO_2_ and V_2_O_5_ nanowires [14]. Additionally, nanocrystals of another manganese oxide, Mn_3_O_4_, have also been found to exhibit an effect similar to the action of the above-mentioned three enzymes (SOD, catalase, and glutathione peroxidase) [15] and have been reported to be highly effective in reactive oxygen species (ROS) elimination (in vitro and in vivo), with the ability to reduce ear inflammation in laboratory mice [16]. Importantly, poly(acrylic) acid-coated Mn_3_O_4_ nanoparticles (NPs) have been demonstrated to attenuate acute liver injury by inhibiting ferroptosis, a type of iron-driven cell death critically dependent on ROS accumulation and subsequent lipid peroxidation [17].

Thus, Mn_3_O_4_ nanocrystals could be potentially pharmacologically utilized as redox balance and inflammation modulators. However, although manganese oxide nanomaterials are reported to elicit beneficial, therapeutically exploitable effects, their toxicity is also reported to be mediated by cell death induction (e.g., apoptosis) [18,19,20]. Since the eryptosis of mature erythrocytes is closely related to the apoptosis of nucleated cells [12,21], in this study, we evaluated the hemocompatibility of Mn_3_O_4_ nanocrystals as a step in the assessment of their overall biocompatibility and safety, focusing on eryptosis and molecular mechanisms of their erythrotoxicity.

## 2. Results

### 2.1. Characterization for Mn_3_O_4_ Nanoparticles

Transmission electron microscopy (TEM) images of the synthesized Mn_3_O_4_ NPs revealed a rhombus-like morphology of the nanocrystals, with an edge length of approximately 16 nm (Figure 1A). The energy-dispersive X-ray spectroscopy (EDS) analysis of the energy-dispersive X-ray spectrum correctly identified the presence of Mn and O elements (Figure 1B).

The crystal structure of the synthesized Mn_3_O_4_ NPs was further analyzed using powder X-ray diffraction (PXRD, Figure 1C). Based on the PDF-5^+^ 2024 International Center for Diffraction Data ICDD-database (ICDD, https://www.icdd.com, accessed on 27 March 2025), the identified phase corresponded to Mn_3_O_4_ (PDF card: 00-024-0734.cif) with a tetragonal crystal lattice. According to Rietveld refinement calculations, the lattice parameters of the nanocrystal were as follows: a = b = 5.7 Å; c = 9.47 Å; α = β = γ = 90°, with a unit cell volume of 314.5 ${Å}^{3}$. The calculated crystallite size was 14.6 nm, which was consistent with the TEM data.

According to the high-resolution transmission electron microscopy (HR-TEM) images (Figure 2), the synthesized Mn_3_O_4_ nanocrystals could be ascribed as quadrilateral prisms with a rhombic base and rectangular side faces. Analysis of the interplanar spacings present in the images indicated that the rhombic base of the prism had a densely packed structure (Figure 2A). This base was primarily formed by (013) crystal planes, with an interplanar spacing of approximately 0.28 nm (Figure 2B).

In contrast to the densely packed rhombic base, the rectangular side faces of the prism were less tightly packed and formed by (011) crystal planes (Figure 2C). The interplanar spacing corresponding to these planes on the surface of the prism’s side faces was approximately 0.49 nm (Figure 2D).

The synthesized Mn_3_O_4_ NPs exhibited a negative surface charge due to the stabilizer, with a ζ-potential of—26 ± 12 mV. The hydrodynamic diameter (d_h_) was determined to be 60.6 ± 0.4 nm. The larger d_h_ value compared to the crystallite size obtained via the Rietveld refinement was attributed to the presence of a hydration shell surrounding the NPs in water.

The synthesized solutions of Mn_3_O_4_ NPs were characterized by low levels of metal impurities (Appendix A).

### 2.2. Mn_3_O_4_ Nanoparticles Induce Hemolysis Only at High Concentrations and Do Not Affect Osmotic Fragility

The complex analysis of Mn_3_O_4_ NP hemocompatibility included an assessment of Mn_3_O_4_ NP-induced hemolysis and eryptosis and the impact on the osmotic fragility of erythrocytes. Mn_3_O_4_ NPs did not trigger spontaneous hemolysis at concentrations of up to 40 mg/L (Figure 3A). At the same time, the highest tested concentration of Mn_3_O_4_ NPs (80 mg/L) increased the amount of hemolyzed erythrocytes only approximately twice compared to the non-exposed controls. In line with the low hemolytic activity, Mn_3_O_4_ NPs did not reduce the osmotic fragility of erythrocytes (Figure 3B).

### 2.3. Mn_3_O_4_ Nanoparticles Induce Eryptosis-Associated Phosphatidylserine Externalization at Lower Concentrations than Hemolysis

The cytotoxicity of Mn_3_O_4_ NPs against erythrocytes was additionally tested by the determination of phosphatidylserine expression on the surface of the erythrocytes, the most characteristic feature of eryptosis. As shown in Figure 3C,D, Mn_3_O_4_ NPs triggered substantial eryptosis at concentrations of 40 and 80 mg/L. Therefore, these two concentrations were selected for further assessment of the underlying molecular mechanisms. To further exclude the impact of heavy metal impurities on the outcome of the eryptosis assays, erythrocytes were incubated with Mn_3_O_4_ NPs at high concentrations in the presence/absence of the heavy metal chelators deferoxamine (0.1 mM) and neocuproine (0.1 mM), which can bind iron and copper, respectively. As demonstrated in Appendix A, the presence of chelators did affect the degree of eryptosis, suggesting that iron and copper impurities were not responsible for the Mn_3_O_4_ NP-induced eryptosis.

### 2.4. Mn_3_O_4_ Nanoparticles Induce Ca^2+^ Overload in Mature Erythrocytes

Ca^2+^ signaling is crucial for the execution of eryptosis. The single-cell-free calcium-specific fluorescence intensity of the erythrocytes exposed to Mn_3_O_4_ NPs was found to complement the phosphatidylserine externalization analysis and verified the occurrence of eryptosis. Indeed, Mn_3_O_4_ NPs promoted dose-dependent Ca^2+^ overload in erythrocytes (Figure 4A and Figure 5).

### 2.5. ROS and RNS Contribute to Mn_3_O_4_ Nanoparticle-Induced Eryptosis

Oxidative stress has been widely reported as a trigger of eryptosis; therefore, we assessed the Mn_3_O_4_ NP-triggered production and accumulation of ROS and RNS as markers of oxidative stress. As shown in Figure 4B and Figure 6, ROS production increased with an increase in the concentration of the Mn_3_O_4_ nanocrystals. Likewise, Mn_3_O_4_ NP-treated red blood cells showed a dose-dependent elevation of RNS production (Figure 4C and Figure 7). Thus, oxidative stress at least partially contributed to the Mn_3_O_4_ NP-induced eryptosis.

### 2.6. Mn_3_O_4_ Nanoparticle-Induced Eryptosis Is Associated with an Altered Lipid Order of Cell Membranes

Oxidative stress is known to promote lipid peroxidation, associated with the alteration of the lipid order of cell membranes. To study the effect of Mn_3_O_4_ NPs on the cell membrane of erythrocytes, we used the NR12S probe, a widely applied fluorescent probe utilized to monitor changes in the lipid order of cell membranes. As shown in Figure 4D and Figure 7, Mn_3_O_4_ NPs caused a red shift in the emission spectrum of the NR12S probe, suggesting a decrease in the lipid order. Notably, a higher concentration (80 mg/L) of Mn_3_O_4_ NPs promoted a more pronounced red shift (*p* < 0.05) than the lower one (40 mg/L). Thus, Mn_3_O_4_ NPs reduced the lipid order dose-dependently.

### 2.7. Mn_3_O_4_ Nanoparticles Trigger Calpain Activation in Mature Erythrocytes

Calpain is a calcium-activated protease that mediates cytoskeleton degradation during eryptosis. As shown in Figure 8, Mn_3_O_4_ NPs activated calpain in a dose-dependent manner.

### 2.8. The Extrinsic Pathway at Least Partly Contributes to Mn_3_O_4_ Nanoparticle-Induced Eryptosis

It is important to note that caspases are not indispensable for the execution of eryptosis. However, they are recruited within the Fas-dependent extrinsic eryptotic pathway. Therefore, their activity was analyzed to evaluate the ability of Mn_3_O_4_ nanocrystals to induce caspase-dependent eryptosis. As illustrated in Figure 4E,F and Figure 9, caspase-3 as well as caspase-8 were activated in erythrocytes exposed to Mn_3_O_4_ NPs. Caspase-8 recruitment clearly suggested that the activation of the extrinsic pathway mediated Mn_3_O_4_ NP-induced eryptosis.

### 2.9. Regulated Cell Death Triggered by Mn_3_O_4_ Nanoparticles Is Aggravated by Ascorbic Acid and Is Not Associated with RIPK1 Recruitment

To further investigate the contribution of ROS to Mn_3_O_4_ nanocrystal-induced eryptosis, incubation with ascorbic acid was applied. Unexpectedly, ascorbic acid exacerbated Mn_3_O_4_ nanocrystal-induced cell death (Figure 10A).

The involvement of receptor-interacting serine/threonine-protein kinase 1 (RIPK1) in Mn_3_O_4_ nanocrystal-triggered cell death was assessed by incubation in the presence of necrostatin-1. Necrostatin-1 did not affect the proportion of eryptotic cells; therefore, RIPK1 was probably not recruited or involved in the process (Figure 10B).

## 3. Discussion

Herein, we assessed the cytotoxicity of Mn_3_O_4_ NPs against mature erythrocytes, with a particular focus on the evaluation of eryptosis parameters. We identified several molecular events and pathways involved in the execution of Mn_3_O_4_ nanocrystal-induced erythrocyte cell death. Mn_3_O_4_ nanocrystals did not show any erythrotoxicity up to the concentrations of at least 20 mg/L. Erythrocytes seem to be less sensitive to Mn_3_O_4_ NP cytotoxicity compared to other cell types reported in other studies, but comparison between different studies can be challenging due to the varying physical and chemical properties of synthesized Mn_3_O_4_ nanocrystals. In particular, Mn_3_O_4_ NPs triggered apoptosis and ROS production in pheochromocytoma PC12 cells at 5 mg/L [19]. Fernández-Pampín et al. reported that the ROS-dependent cell death of A549 human alveolar basal epithelial cells and HT-29 human colon cancer cells was induced by Mn_3_O_4_ nanocrystals at 10 mg/L [20]. Additionally, Mn_3_O_4_ NPs promoted the cell death of MCF7 breast cancer cells starting from 10 mg/L [22]. Of note, the majority of studies, including those mentioned above, indicate that Mn_3_O_4_ NP-mediated cytotoxicity is linked to ROS overgeneration and subsequent apoptosis. Our study supports these observations, demonstrating that Mn_3_O_4_ nanocrystals trigger eryptosis (phosphatidylserine externalization and Ca^2+^ overload). At the same time, Mn_3_O_4_ NP-induced eryptosis is associated with ROS and RNS accumulation. ROS are known to be important messengers in eryptosis, contributing to Ca^2+^ influx [23,24]. Although much less is known about the contribution of RNS to eryptosis, it is clearly demonstrated that RNS are involved in apoptosis [25]. We have documented a substantial induction of oxidative stress in the erythrocytes exposed to Mn_3_O_4_ nanocrystals, which might be associated with the activation of lipid peroxidation and consequent lipid order disruption. Indeed, we detected a Mn_3_O_4_ NP-induced decrease in the lipid order of cell membranes, which might be indicative of lipid peroxidation [26]. Nevertheless, ascorbic acid aggravated Mn_3_O_4_ nanocrystal-induced eryptosis, suggesting that the execution of eryptosis in this case did not require ROS signaling. These findings are consistent with other studies showing that ascorbic acid might strengthen aluminum-induced [27] and chemerin-triggered eryptosis [28].

Ca^2+^ signaling is a known master regulator of eryptosis. Ca^2+^ ions mediate eryptosis-linked cell shrinkage through the Gardos effect and membrane vesiculation via calpain activation [29]. Calpain is a Ca^2+^-dependent protease responsible for the disintegration of the cytoskeleton. Its activation determines eryptosis-associated membrane blebbing [10]. It is important to note that Mn_3_O_4_ nanocrystals dose-dependently trigger an increase in intracellular Ca^2+^ in nucleated PC12 cells as well [19]. However, the authors emphasize the importance of mitochondrial Ca^2+^ overload in Mn_3_O_4_ NP-triggered apoptosis. Along with calpain, there are other proteolytic caspases that are implicated in Mn_3_O_4_ nanocrystal-induced eryptosis. Our study demonstrates that the only two caspases retained in mature erythrocytes, caspase-3 and caspase-8 [21,30,31], are activated by Mn_3_O_4_ nanocrystals. Notably, compelling evidence suggests that caspases are non-essential for the execution of eryptosis. This might be one of the major differences between eryptosis and apoptosis since apoptosis is defined as a type of caspase-dependent cell death [11,12]. However, erythrocytes were shown to express Fas, and Fas signaling in erythrocytes promoted phosphatidylserine externalization through the recruitment of caspase-8 and caspase-3 [32]. Further studies consequently confirmed the existence of Fas/caspase-8/caspase-3 pathway-dependent eryptosis [33,34]. Thus, caspase-8 recruitment in Mn_3_O_4_ nanocrystal-induced eryptosis suggests that the extrinsic eryptotic pathway is triggered by the studied nanomaterials.

Erythrocytes have been shown to undergo necroptosis mediated by RIPK1. There is some evidence that erythrocyte necroptosis and eryptosis are mutually exclusive [21,35]. Our findings indicate that Mn_3_O_4_ NP-triggered eryptosis does not require RIPK1 recruitment. This suggests that Mn_3_O_4_ NPs do not trigger the necroptosis of erythrocytes. It can be assumed that caspase-8 activation might switch a cell death decision towards eryptosis [21,35].

Our findings can provide insight into the clinical manifestations of Mn_3_O_4_ nanocrystal-induced toxicity. Indeed, since eryptotic cells are readily cleared from circulation, the acceleration of eryptosis is linked to anemia [36]. Moreover, eryptotic cells interact with platelets to increase blood clotting [33]. Eryptosis-associated endothelial damage might be attributable to the phosphatidylserine-mediated CXC chemokine ligand 16 (CXCL16)-dependent adherence of eryptotic cells to endothelial cells [37]. At the same time, this study indicates that nanomaterials are potential eryptosis-targeting nanodrugs. It can be admitted that further research focused on nanomaterial-induced eryptosis can help unearth the novel aspects of nanomaterial-mediated hematotoxicity as a whole and erythrotoxicity in particular. Currently, it is poorly studied how the various physical and chemical characteristics of nanomaterials (i.e., composition features, size, shape, charge, doping, etc.) might influence their toxicity against erythrocytes, which is an important issue in nanotoxicology and nanodrug discovery.

## 4. Materials and Methods

### 4.1. Chemicals and Reagents

Manganese (II) chloride MnCl_2_·4H_2_O (Sigma-Aldrich, Burlington, MA, USA, >99%), sodium hydroxide NaOH (Sigma-Aldrich, >95%), and tri-sodium citrate NaCt·5.5 H_2_O (Merck, Darmstadt, Germany, >99%) were used as received. DMSO (D12345) was supplied by Invitrogen (Waltham, MA, USA). DPBS (14190144) and PBS buffers were purchased from Gibco (Waltham, MA, USA), while NP-40 buffer (J60766.AP) was provided by Thermo Scientific (Waltham, MA, USA). Necrostatin-1 was purchased from Apollo Scientific (Bredbury, UK). The following reagents were used for the detection of eryptosis markers: Annexin V-FITC (BD Pharmingen™, FITC annexin V, Franklin Lakes, NJ, USA), Fura 2-AM (Abcam, Cambridge, UK, ab120873), Cellular ROS/RNS Detection Assay Kit (Abcam, ab139473), NR12S probe (Cytoskeleton Inc., Denver, CO, USA, MG08), Caspase Multiplex Activity Assay Fluorometric Kit (Abcam, ab219915), and Calpain Activity Assay Kit (Abcam, ab65308). Deferoxamine (HY-B1625, purity 99.2%) and neocuproine (HY-W004563, purity 99.82%) were provided by MedChemExpress (Monmouth Junction, NJ, USA).

### 4.2. Synthetic Route for Mn_3_O_4_ Nanoparticles

Stable colloidal solutions of Mn_3_O_4_ NPs were synthesized via the modified wet chemical synthesis route. The procedure comprised the following steps: first, 100 mL of 0.005 M MnCl_2_ solution was mixed with 8 mL of 0.1 M NaOH solution in a 200 mL chemical beaker. The mixture was stirred continuously using a magnetic stirrer for 1 h to facilitate the formation of a precipitate. The resulting precipitate was isolated via centrifugation at 7000 rpm for 15 min. Thereafter, the separated precipitate underwent a series of washing steps to remove impurities. Each washing cycle involved the addition of 100 mL of distilled water to the precipitate, thorough mixing, and subsequent centrifugation at 7000 rpm for 15 min. This process was repeated three times. After the washing step, the precipitate was transferred to a round-bottom flask equipped with a reflux condenser with 50 mL of distilled water added. The suspension was boiled for 10 h. In the final stage, the dark-brown precipitate was again separated by centrifugation at 7000 rpm for 15 min. This process was repeated three times. The precipitate was separated and 100 mL of distilled water was added. The solution was treated with ultrasound at a frequency of 22 kHz for 5 min. Then, 5 mL of a 0.1M sodium citrate solution was added and the solution was again treated with ultrasound at a frequency of 22 kHz. The resulting dark brown colloidal solution was placed in a round-bottom flask and refluxed for 10 h. This solution was filtered through an ultrafiltration membrane with a pore size of 0.45 µm. The pH of the final solution was determined to be 7.1–7.5, with a solid-phase concentration of 165 mg/L.

### 4.3. Characterization of the Synthesized Mn_3_O_4_ Nanoparticles

Mn_3_O_4_ NPs were characterized using high-resolution transmission electron microscopy (HR-TEM). Analysis was conducted on a JEOL JEM-2100F TEM equipped with a Schottky-type field emission gun operated at an accelerating voltage of 200 kV. TEM imaging was performed with a magnification of up to 1,000,000× and selected area electron diffraction (SAED) was used for structural analysis. Images and SAED patterns were captured using Gatan Ultrascan 1000 and Gatan Orius 200D cameras. The chemical composition of the NPs was determined via energy-dispersive X-ray spectroscopy (EDS). The samples were prepared using Holey Carbon/Continuous Ultrathin carbon film Supported Copper Grids (Sigma-Aldrich, size 300 mesh, box of 25×). High-resolution TEM images were processed with Gatan DigitalMicrograph v3.3 software to extract interplanar spacing (d-spacing) values from the crystalline structures of individual NPs.

The phase composition of the Mn_3_O_4_ NPs was determined using powder X-ray diffraction (PXRD) analysis. Measurements were conducted with a Bruker D8 Discover powder diffractometer equipped with Cu Kα radiation and configured in reflection geometry (Bragg–Brentano mode). The instrument featured automatic divergence slits and a motorized beam knife to minimize the background noise caused by air scattering. For sample preparation, a zero-background holder composed of a Si wafer was employed to ensure optimal data quality.

Phase identification was carried out using the PDF-5^+^ 2024 International Center for Diffraction Data ICDD-database (ICDD, https://www.icdd.com, accessed on 27 March 2025), the crystallite size was evaluated through Rietveld refinement using HighScore Plus crystallographic software, with a Si standard employed for calibration and accuracy.

Zeta potentials (ζ-potentials) were estimated using a ZetaPALS/BI-MAS analyzer (Brookhaven Instruments Corp., Nashua, NH, USA).

To assess metal contaminants in solutions of Mn_3_O_4_ NPs, optical emission spectroscopy with inductively coupled plasma (ICP-OES) was carried out using an iCAP6300 Duo spectrometer (Thermo Fisher Scientific, Waltham, MA, USA).

### 4.4. Detection of Hemolysis and the Osmotic Fragility of Erythrocytes

Blood was collected from the caudal vein of intact adult male WAG rats. An aliquot of freshly collected blood was used to prepare an erythrocyte suspension with a total hematocrit of 0.4% in the Ringer solution (125 mM NaCl, 5 mM KCl, 2 mM CaCl_2_, 2 mM MgCl_2_, 32 mM HEPES, 5 mM glucose). The cells were incubated at 5% CO_2_ for 24 h with Mn_3_O_4_ NPs at 0–0.31–0.62–1.25–2.5–5–10–20–40–80 g/L. After the incubation, hemolysis parameters, osmotic fragility, and phosphatidylserine externalization were assessed. For the hemolysis assay, the cells after incubation with the Mn_3_O_4_ NPs were centrifuged at 500× *g* for 5 min. The content of hemoglobin in the supernatant was identified using a ULAB 102UV spectrophotometer (ULAB, Nanjing, China) at 541 nm. To analyze the osmotic fragility, the cells incubated with the NPs were washed and resuspended in 1 mL 0.3% NaCl for 10 min at 37 °C. Thereafter, the samples were centrifuged at 500× *g* for 5 min and hemoglobin in the supernatant was quantified with a ULAB 102UV spectrophotometer (ULAB, China) at a wavelength of 541 nm.

### 4.5. Detection of Eryptosis by Analyzing Phosphatidylserine Externalization

To detect phosphatidylserine externalization, the incubated erythrocytes were washed from the NPs and resuspended in 100 µL annexin-binding buffer with Annexin V-FITC. The cells were incubated with Annexin V-FITC for 15 min with no exposure to light [38]. Then, the cells were washed and resuspended in 100 µL Annexin-binding buffer for further measurements. The fluorescence was collected in the FL1 channel (FITC, excitation 488 nm and emission 525 nm) using a BD FACS Canto™ II flow cytometer (Becton Dickinson, Franklin Lakes, NJ, USA). Data were processed by FlowJo™ (v10, BD Biosciences, Franklin Lakes, NJ, USA) software.

### 4.6. Confocal Microscopy-Based Detection of Intraerythrocytic Ca^2+^ Levels

The intracellular content of calcium ions was imaged directly using a Fura 2-AM probe. Briefly, to stain erythrocytes with the Fura 2-AM probe, an appropriate aliquot of its stock solution in DMSO was added to 0.25 mL of DPBS buffer. The mixtures were slightly vortexed (IKA TTS3 shaker) and the obtained solution was at once added to 0.5 mL of the RBC suspension. Thus, the final concentration of the Ca^2+^-sensitive probe was 10 μM (<0.25% DMSO). Prior to data acquisition, the suspensions of erythrocytes were incubated with the probe for 1 h at 37 °C in the dark using the Galaxy 14S incubator (Eppendorf). The images were acquired by a confocal microscope (excitation 350 nm/emission 505 nm) as described below [39]. After image processing, which included background subtraction, deconvolution, and image segmentation to identify intracellular regions of interest, a qualitative measurement of calcium was performed. Fura 2-AM-Ca^2+^-specific fluorescence changes were expressed in relative fluorescence units (rfus) for each sample and calculated as relative fluorescence units/cell.

### 4.7. Detection of ROS and RNS in Erythrocytes

Total intracellular levels of ROS and RNS were detected directly using a dual Cellular ROS/RNS Detection Assay Kit. ROS- and RNS-specific non-fluorescent, cell-permeable ROS Detection Reagent (Green) and NO Detection Reagent (Red) dyes were added to 0.5 mL of pre-warmed (37 °C) DPBS buffer. After vortexing (IKA TTS3 shaker, 4 s), the dye-containing solutions were added to RBC suspensions (0.5 mL), reaching a final concentration of 0.01 μM. The erythrocytes were incubated for 30 min at 37 °C, avoiding exposure to light (Galaxy 14S incubator). The images reflecting the content of ROS (excitation 490 nm/emission 525 nm) and RNS (excitation 650 nm/emission 670 nm) in red blood cells were captured by a confocal microscope as described below.

### 4.8. NR12S Probe-Based Detection of the Lipid Order in Cell Membranes of Erythrocytes

A Nile Red dye-based ratiometric NR12S probe was used to identify modifications of the lipid order in cell membranes following exposure to Mn_3_O_4_ NPs. The fluorescence emission spectra of this probe reflected the changes in the lipid order. Shifts towards shorter wavelengths indicated incorporation into a liquid-ordered phase in comparison with a liquid-disordered phase. The emission ratio (633 nm/550 nm) of the probe was used to identify changes in the lipid order of phospholipid bilayers in the cell membrane [40].

To stain erythrocytes exposed to Mn_3_O_4_ NPs with the NR12S probe, 0.01 μM working solutions were prepared by dissolving an appropriate aliquot of the NS12 probe stock solution in DMSO in 0.5 mL of DPBS buffer. The stained erythrocytes were incubated for 7 min at room temperature in the dark [41].

To quantify the changes in the lipid order, the ratiometric index R, which is the liquid-disordered phase-specific fluorescence intensity (633 nm, “red” channel)/liquid-ordered phase-specific fluorescence intensity (550 nm, “green” channel) ratio, was calculated based on confocal microscopy-acquired data, as described below.

### 4.9. Confocal Microscopy-Based Evaluation of Extrinsic Eryptosis by Detecting the Activity of Caspase-3 and Caspase-8

The activity of cleaved caspase-3 and caspase-8 was measured directly in erythrocytes exposed to Mn_3_O_4_ NPs by a Caspase Multiplex Activity Assay Fluorometric Kit. The kit enables the fluorogenic indicator-based (DEVD-ProRed and IETD-R110 for caspase-3 and caspase-8, respectively) quantification of the activity of caspases by releasing ProRed™ (red fluorescence) and R110™ (green fluorescence) fluorophores upon the activation of caspases by limited proteolysis. The fluorescence of ProRed™ and R110™ reflected the activity of the corresponding caspases. The caspase-specific DEVD-ProRed and IETD-R110 reagents were placed into 0.5 mL of pre-warmed (37 °C) assay buffer. The mixtures were vortexed (IKA TTS3 shaker, 4 s) and added to 0.5 mL of the RBC suspensions to reach a final concentration of 0.01 μM. Incubation lasted for 30 min at 37 °C and the cells were kept in the dark (Galaxy 14S incubator). Confocal microscopy at an excitation of 535 nm and emission of 620 nm (caspase-3, “red” channel) and an excitation of 650 nm and emission of 670 nm (caspase-8, “green” channel) was used to acquire the fluorescence, as described below.

### 4.10. Confocal Microscopy

A laser scanning confocal microscope FV10i-LIV (Olympus, Japan) equipped with a 60/1.2 NA water immersion objective was employed to visualize erythrocytes and acquire the fluorescence. Erythrocytes were placed into single-well PTFE slides (Thermo Scientific, cat. no. X2XER203B#, 13 µL cell suspension/well), which were covered with coverslips (Ibidi, cat. no. 10812). Imaging was carried out in a scanning mode format of 1024 × 1024 pixels, with a pinhole aperture of 1 Airy unit. To ensure the representativeness of our data, six fields of view in different regions of single-well PTFE slides were analyzed. Fifty cells per sample were scored. Post-rendering of the obtained images, which included autofluorescence subtraction, measurements of fluorescence intensities and ratiometric analyses were performed with the help of Olympus cellSens software v.1.18 (Olympus licensed).

### 4.11. Detection of Calpain Activity

The calpain activity in erythrocytes treated with Mn_3_O_4_ NPs was performed using a Calpain Activity Assay Kit. The cells were washed twice and resuspended in PBS (pH 7.4). The cell number was determined using an automatic cell counter Scepter 2.0 (Millipore, Burlington, MA, USA) and Scepter Software Pro 2.1 (Millipore). NP-40 buffer was used to lyse the erythrocytes. The native substrate Ac-LLY-AFC emitted blue light (emission 400 nm) and, upon its cleavage bycalpain,-free AFC was formed, which emitted a yellow-green fluorescence (emission 505 nm). The 505/400 nm emission ratio was calculated to quantify the calpain activity. The measurements were performed using an FL-600 multimodal microplate reader (Bio-Tek, Winooski, VT, USA). Determination of the initial velocity for the calpain activity was calculated in the linear region of the ratiometric curve.

### 4.12. Detection of the Mechanisms of Mn_3_O_4_ Nanoparticle-Induced Eryptosis

To further analyze the mechanisms involved in Mn_3_O_4_ NP-induced eryptosis, these NPs at the concentrations triggering eryptosis (i.e., 40 and 80 mg/L) were incubated in the presence of ascorbic acid (0.5 mM) or necrostatin-1 (10 µM) for 24 h at 5% CO_2_. Thereafter, the cells incubated with Mn_3_O_4_ NPs in the presence or absence of the abovementioned inhibitors were stained with Annexin V-FITC, as reported above. Furthermore, to exclude the possible effects of heavy metal impurities, erythrocytes were incubated with Mn_3_O_4_ NPs at the concentrations triggering eryptosis (40 and 80 mg/L) in the presence/absence of the heavy metal chelators deferoxamine (0.1 mM) and neocuproine (0.1 mM), which are capable of binding iron and copper, respectively. Annexin V-FITC staining was performed as described above to identify the degree of eryptosis.

### 4.13. Statistical Analysis

Statistical differences were determined by one-way analysis of variance (ANOVA) followed by Tukey’s test for multiple comparisons using Graph Pad Prism 5.0 software (USA). If two independent variables were compared, the *t* test was used. The difference was considered statistically significant at *p* < 0.05. The mean ± standard deviation (SD) was calculated.

## 5. Conclusions

Our findings indicate that Mn_3_O_4_ nanocrystals are hemocompatible and promote the cell death of erythrocytes at concentrations of 40 mg/L. The Mn_3_O_4_ nanocrystal-induced cell death of erythrocytes occurred in the form of eryptosis featuring phosphatidylserine externalization, Ca^2+^ overload, reactive oxygen species and reactive nitrogen species accumulation, a decrease in the lipid order of the cell membrane, and the activation of calpain, caspase-3, and caspase-8. This study provides additional evidence that eryptosis evaluation is a promising tool in nanotoxicological research and confirms that eryptosis has a higher sensitivity in comparison to hemolysis testing and osmotic fragility assessment. This pre-clinical in vitro research expands our knowledge of the toxicological profile of manganese oxide nanoparticles, preparing the way for their further clinical evaluation as pharmaceutical agents.

## Figures and Tables

**Figure 1 ijms-26-03284-f001:**
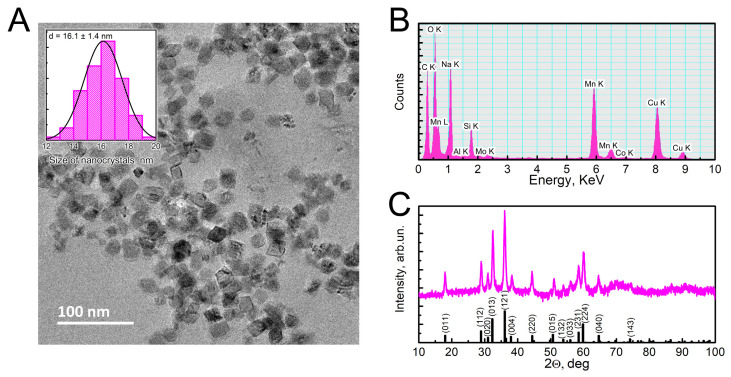
Characterization of synthesized Mn_3_O_4_ nanoparticles using transmission electron microscopy (TEM), energy-dispersive X-ray spectroscopy (EDS), and powder X-ray diffraction (PXRD). Synthesized Mn_3_O_4_ nanocrystals had rhombus-like morphology (panel (**A**), TEM image including size distribution histogram), their spectrum correctly included Mn and O elements (energy-dispersive X-ray spectroscopy, EDS, panel (**B**)), and the identified phase corresponded to Mn_3_O_4_ (XRD, panel (**C**)).

**Figure 2 ijms-26-03284-f002:**
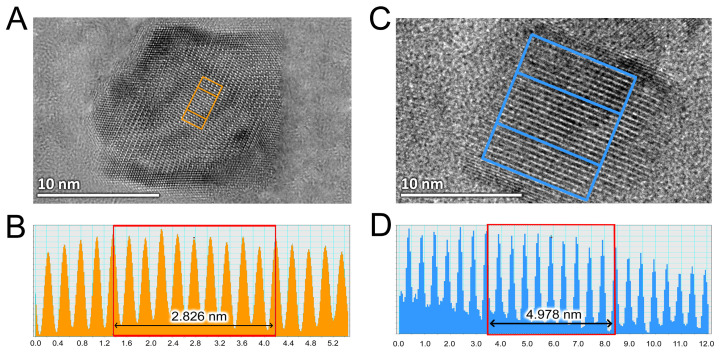
Characterization of Mn_3_O_4_ nanocrystals with high-resolution transmission electron microscopy (HR-TEM). HR-TEM images of the rhombic base face (panel (**A**)) with a profile-view image (panel (**B**)) and the rectangular side face (panel (**C**)) with a profile-view image (panel (**D**)) for the synthesized Mn_3_O_4_ nanocrystals.

**Figure 3 ijms-26-03284-f003:**
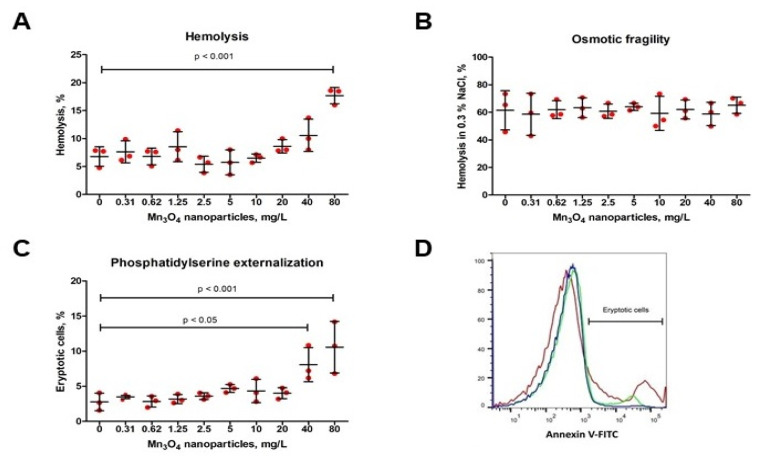
Mn_3_O_4_ NPs show good hemocompatibility: hemolysis assay (panel (**A**)) and osmotic fragility assay (panel (**B**)). Flow cytometry-based annexin V-FITC staining: identification of the percentage of phosphatidylserine-displaying eryptotic cells (panel (**C**)). Representative histograms show the percentage of eryptotic cells in the erythrocyte samples treated with Mn_3_O_4_ NPs (0 mg/L—blue line; 40 mg/L—green line; 80 mg/L—red line) for 24 h (panel (**D**)). ANOVA and Tukey tests, mean ± SD, n = 3.

**Figure 4 ijms-26-03284-f004:**
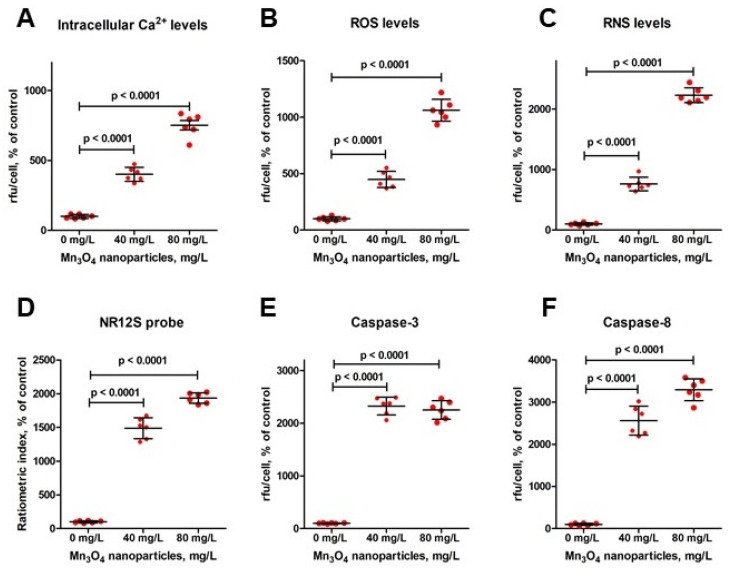
Quantitative determination of eryptosis parameters in erythrocytes incubated for 2 h with Mn_3_O_4_ nanoparticles at 0 mg/L (control samples), 40 mg/L, and 80 mg/L: intracellular calcium levels (panel (**A**)); reactive oxygen species, ROS (panel (**B**)); reactive nitrogen species, RNS (panel (**C**)); the single-cell ratiometric index, i.e., the ratio of the liquid-disordered phase-specific fluorescence intensity Fl red 633 nm to the liquid-ordered phase-specific fluorescence intensity Fl green 550 nm (panel (**D**)); caspase-3 (panel (**E**)); and caspase-8 (panel (**F**)). ANOVA and Tukey test, mean ± SD, n = 6. Note: rfu—relative fluorescence units.

**Figure 5 ijms-26-03284-f005:**
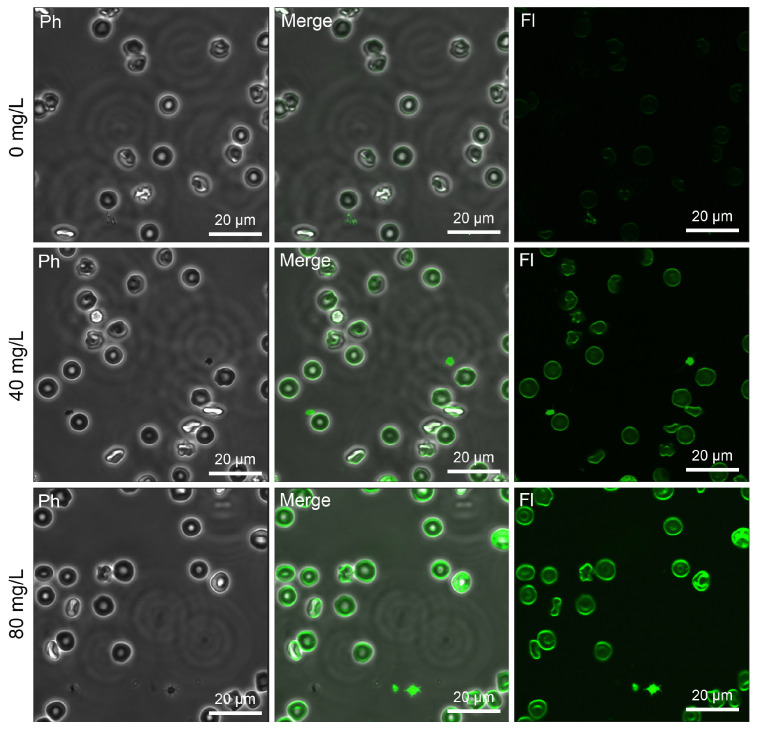
Laser scanning confocal microscopy-based detection of the calcium-specific fluorescence in erythrocytes treated for 2 h with Mn_3_O_4_ nanoparticles at 0 mg/L (control samples), 40 mg/L, and 80 mg/L. Merging combined the fluorescence channel with the phase contrast image. Scale bar is 20 µm. Note: Ph—phase contrast mode, Fl—a fluorescence mode reflecting the calcium-specific fluorescence intensity.

**Figure 6 ijms-26-03284-f006:**
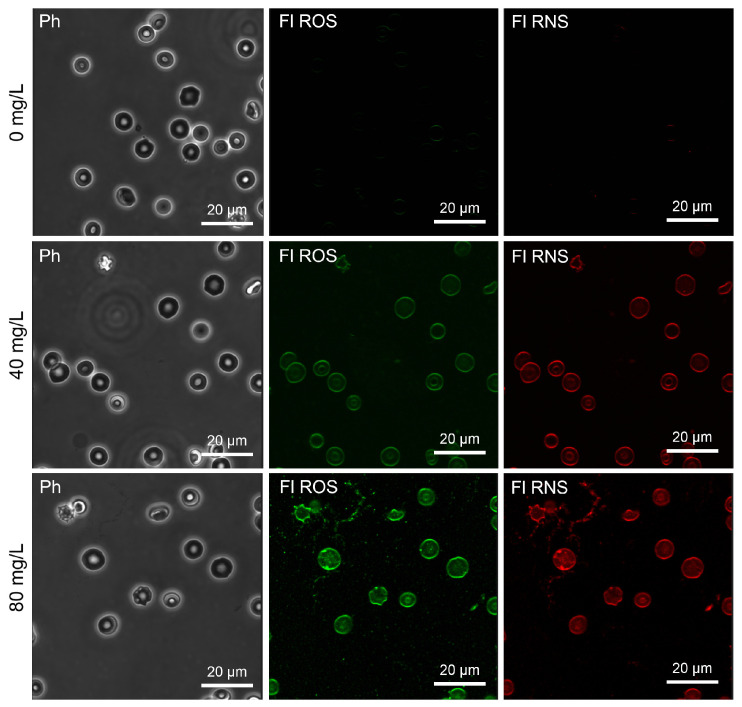
Laser scanning confocal microscopy: visualization of erythrocytes with the reactive oxygen species (ROS)- and reactive nitrogen species (RNS)-specific fluorescence exposed for 2 h of incubation to Mn_3_O_4_ nanoparticles (0–40–80 mg/L). Merging combined the fluorescence channels with the phase contrast image. Scale bar is 20 µm. Note: Ph—phase contrast mode, Fl ROS—a fluorescence mode indicating the ROS-specific fluorescence intensity, Fl RNS—a fluorescence mode demonstrating the RNS-specific fluorescence.

**Figure 7 ijms-26-03284-f007:**
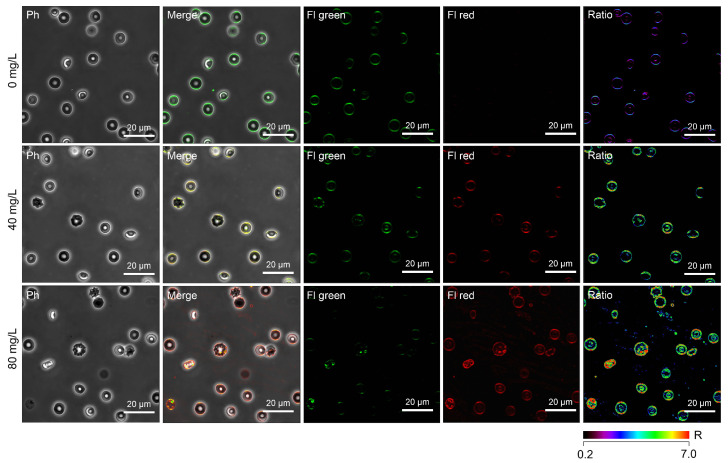
Laser scanning confocal microscopy-based analysis of the lipid order changes in the cell membranes of erythrocytes treated for 2 h with Mn_3_O_4_ nanoparticles at 0 mg/L (control samples), 40 mg/L, or 80 mg/L using the NR12S probe. The fluorescence channels were merged with the phase contrast (merge). Ratiometric images (R = Fl red/Fl green) are shown (ratio). Scale bar is 20 µm. Note: Ph—phase contrast mode, Fl green and Fl red—fluorescence modes showing the liquid-ordered and liquid-disordered phases, respectively.

**Figure 8 ijms-26-03284-f008:**
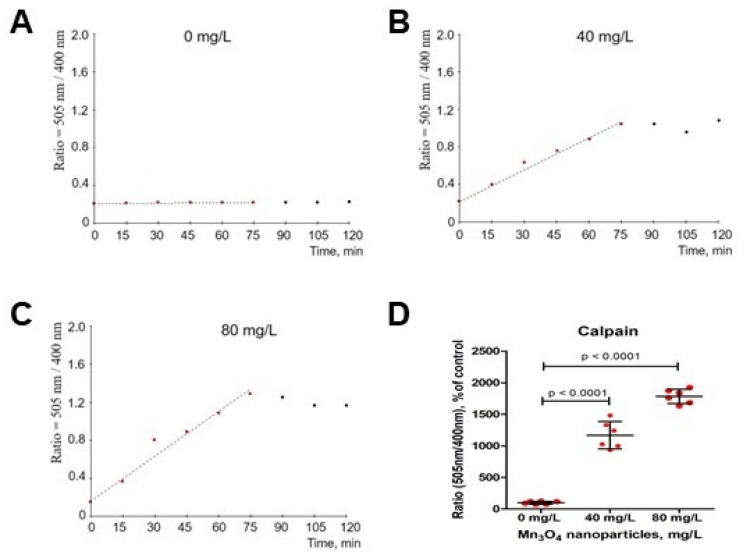
The representative curves representing the calpain activity in erythrocytes exposed to Mn_3_O_4_ nanoparticles at 0 mg/L (panel (**A**)), 40 mg/L (panel (**B**)), and 80 mg/L (panel (**C**)) for 2 h. The calpain activity was quantified by calculating the fluorescence intensity 505 nm (cleavage of the calpain substrate)/400 nm (native substrate) ratio. The calpain activity (velocity, v) was calculated in the linear region of the curve (panel (**D**)). ANOVA and Tukey test, mean ± SD, n = 6.

**Figure 9 ijms-26-03284-f009:**
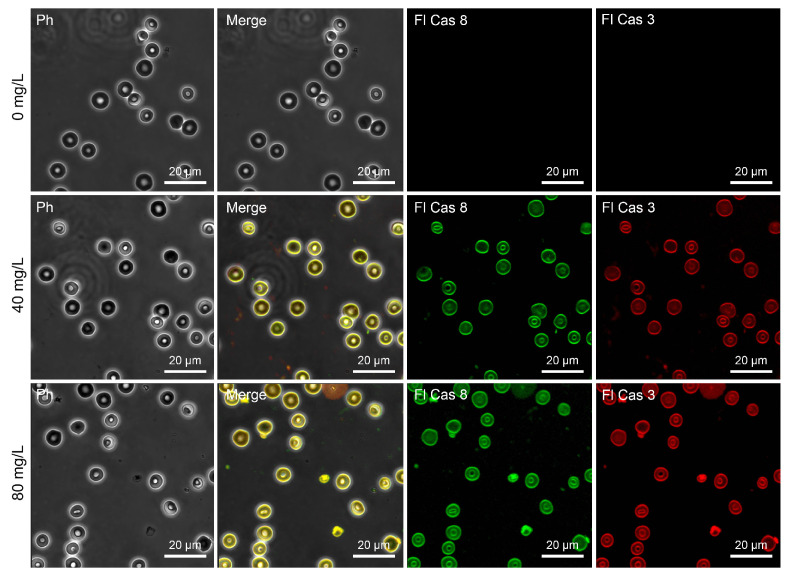
Laser scanning confocal microscopy-based assessment of caspase-dependent eryptosis. Detection of caspase-3- and caspase-8-specific fluorescence in red blood cells incubated for 2 h with Mn_3_O_4_ nanoparticles at 0 mg/L–40 mg/L–80 mg/L. The fluorescence channels were merged with the phase contrast image (merge). Scale bar is 20 µm. Note: Ph—phase contrast mode, Fl Cas 3 and Fl Cas 8—fluorescence modes demonstrating the activity of caspase-3 and caspase-8, respectively.

**Figure 10 ijms-26-03284-f010:**
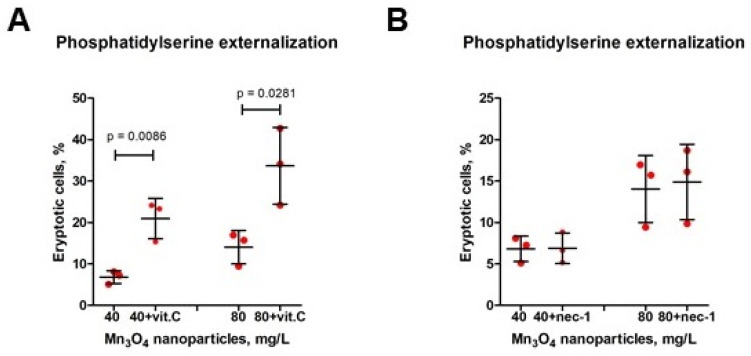
Mn_3_O_4_ nanoparticle-induced cell death is exacerbated by ascorbic acid (vitamin C) and is not mediated by receptor-interacting serine/threonine-protein kinase 1 (RIPK1). Mn_3_O_4_ nanoparticles at 40 mg/L and 80 mg/L were incubated for 24 h in the presence and absence of vitamin C (0.5 mM, panel (**A**)) or necrostatin-1 (10 µM, panel (**B**)) and further stained with Annexin V-FITC to analyze the percentage of eryptosis phosphatidylserine-exposing cells. *T* test, mean ± SD, n = 3.

## Data Availability

The experimental data supporting the findings of this research were generated at V.N. Karazin Kharkiv National University, Institute for Scintillation Materials of the National Academy of Sciences of Ukraine, Kharkiv National Medical University, Stockholm University, and Institute for Problems of Cryobiology and Cryomedicine of the National Academy of Sciences of Ukraine. The data are available from the corresponding authors upon reasonable request.

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
