# Peer review of "Mn3O4 Nanocrystal-Induced Eryptosis Features Ca2+ Overload, ROS and RNS Accumulation, Calpain Activation, Recruitment of Caspases, and Changes in the Lipid Order of Cell Membranes"

_ijms, 2025, doi:10.3390/ijms26073284_

Round 1

Reviewer 1 Report

Comments and Suggestions for Authors

1-      English editing by a professional or native speaker is required.

2-      In the introduction, the connection between Mn3O4 nanocrystals induced eryptosis and the pathways mentioned is somewhat unclear.

3-      A more detailed explanation is required regarding eryptosis relation with haemolysis.

4-      All differences between the groups in results should indicated in percentage.       

5-      Overall, the quality and analysis of all images are quite questionable, which may be considered.

6-      Authors should present their data as means plus/minus S.D. and not standard errors because readers are interested in knowing the dispersion of values and not the precision of the mean due to the paucity of observations.

7-      Add future directions and clinical relevance to the discussion section.

Comments on the Quality of English Language

  English editing by a professional or native speaker is required.

Author Response

  • English editing by a professional or native speaker is required.

Answer: Thank you for pointing this out. We have tried to improve the language quality.

2          In the introduction, the connection between Mn3O4 nanocrystals induced eryptosis and the pathways mentioned is somewhat unclear.

Answer: In the Introduction section, we have summarized the reported effects of manganese oxide nanoparticles to emphasize that they are promising as nanodrugs and their toxicity should be studied. To make the connection smoother, the following paragraph was added.

However, although manganese oxide nanomaterials are reported to elicit beneficial therapeutically exploitable effects, their toxicity is also reported to be mediated by cell death induction (e.g., apoptosis) [18-20]. Since eryptosis of mature erythrocytes is closely related to apoptosis of nucleated cells [12, 21], in this study, we evaluated hemocompatibility of Mn3O4 nanocrystals as a step in the assessment of their overall biocompatibility and safety, focusing on eryptosis and molecular mechanisms of their erythrotoxicity.

3          A more detailed explanation is required regarding eryptosis relation with haemolysis.

Answer: Thank you for pointing this out. Additional explanation was provided.

A higher sensitivity of eryptosis assays compared to hemolysis can be linked to physiological functions of eryptosis. There is accumulating evidence that eryptosis develops in response to external and internal perturbations to erythrocytes, which rapidly culminates in phosphatidylserine externalization and clearance of erythrocytes by macrophages occurring within minutes [10]. Thus, eryptosis of dysfunctional or damaged erythrocytes ensures rapid clearance of these cells before the cell membrane integrity is impaired and the potential detrimental release of the intraerythrocytic content can occur (hemolysis). Admittedly, eryptosis is triggered by subhemolytic lethal stimuli [8].

4          All differences between the groups in results should indicated in percentage.

Answer: Figures were modified. Numerical values were normalized to controls and expressed in %.

5          Overall, the quality and analysis of all images are quite questionable, which may be considered.

Answer: The quality of Figures might be compromised due to their addition to the manuscript file. We have provided Figures of high quality as separate files.

6          Authors should present their data as means plus/minus S.D. and not standard errors because readers are interested in knowing the dispersion of values and not the precision of the mean due to the paucity of observations.

Answer: Thank you for pointing this out. Figures were modified to express numerical values as mean and SD.

7          Add future directions and clinical relevance to the discussion section.

Answer: Thank you. This discussion was added.

Our findings can provide insight into clinical manifestations of Mn3O4 nanocrystals-induced toxicity. Indeed, since eryptotic cells are readily cleared from the circulation, acceleration of eryptosis is linked to anemia [35]. Moreover, eryptotic cells interact with platelets to increase blood clotting [32]. Eryptosis-associated endothelial damage might be attributed to phosphatidylserine-mediated CXC chemokine ligand 16 (CXCL16)-dependent adherence of eryptotic cells to endothelial cells [36]. At the same time, this study indicates that nanomaterials are potential eryptosis-targeting nanodrugs. It can be admitted that further research focused on nanomaterials-induced eryptosis can help unearth novel aspects of nanomaterials-mediated hemotoxicity as a whole and erythrotoxicity in particular. Currently, it is poorly studied how various physical and chemical characteristics of nanomaterials (i.e. composition features, size, shape, charge, doping, etc.) might influence their toxicity against erythrocytes, which is an important issue of nanotoxicology and nanodrug discovery.

We would like to express our sincere gratitude to the Reviewer for informative comments that have helped improve the quality of the manuscript. The corrections are made in the Track Changes mode.

Reviewer 2 Report

Comments and Suggestions for Authors

In this manuscript, Kot and collaborators made a tentative characterization of the Mn3O4 nanocrystals toxicity inducing eryptosis. In the absence of additional data, this toxicity is not sustained by a direct effect of Mn3O4 NPs, but by the presence of metal contaminants in their samples. In this sense, authors require to perform NPs characterization experiments to probe that their toxicity is not induced by metal impurities in their NPs before this manuscript is published.

Major points:

1) A list of chemicals, providers and purity should be included in the Material and methods section.

2) In lines 153-155, the authors stated: “Mn3O4 NPs did not trigger spontaneous hemolysis at concentrations of up to 40 mg/L (Figure 3A)”. Therefore, in the rest of experiments, authors used this concentration and higher to induce eryptosis. 

This is a huge concentration of NPs, and support that toxicity might be associated with other metals present in their composition. The authors need to characterize their nanoparticles with respect to other impurities (metals), for example using GC-MS, and prove that they are not in charge of this toxicity.

3) To confirm that observed toxicity, oxidative stress and cell death parameters are induced by Mn3O4 NPs and not associated with other metal contaminants, the authors should perform experiments using chelators that are probable in charge of this effect (such as iron or cupper). For this purpose, authors are suggested to use deferoxamine or neocuproine or other specific metal/chelators to avoid Fenton/Haber–Weiss reactions that are probably responsible for the NPs toxicity.

4) It is highly recommended to characterize the pathways involved in the oxidative stress induced by NPs. This characterization require identification and use of electron paramagnetic resonance to detect specific fingerprints to dissect the radical pathway in charge of the oxidative stress.

Author Response

In this manuscript, Kot and collaborators made a tentative characterization of the Mn3O4 nanocrystals toxicity inducing eryptosis. In the absence of additional data, this toxicity is not sustained by a direct effect of Mn3O4 NPs, but by the presence of metal contaminants in their samples. In this sense, authors require to perform NPs characterization experiments to probe that their toxicity is not induced by metal impurities in their NPs before this manuscript is published.

Major points:

1) A list of chemicals, providers and purity should be included in the Material and methods section.

Answer: Thank you for this valuable suggestion. The separate subsection entitled “Chemicals and reagents” was added to the manuscript.

Manganese (II) chloride MnCl2 • 4H2O (Sigma-Aldrich, >99%), sodium hydroxide NaOH (Sigma-Aldrich, >95%), and tri-sodium citrate NaCt•5.5 H2O (Merck, >99%) were used as received. DMSO was supplied by Invitrogen (D12345). DPBS (14190144) and PBS buffers were purchased from Gibco, while NP-40 buffer was provided by Thermo Scientific, (J60766.AP). Necrostatin-1 was purchased from Apollo Scientific, Bredbury, UK. The following reagents were used for detection of eryptosis markers: Annexin V-FITC (BD Pharmingen™, FITC annexin V, Franklin Lakes, NJ, USA), Fura 2-AM (Abcam, ab120873), Cellular ROS/RNS Detection Assay Kit (Abcam, ab139473), NR12S probe (Cytoskeleton Inc., MG08), Caspase Multiplex Activity Assay Fluorometric Kit (Abcam, ab219915), and Calpain Activity Assay Kit (Abcam, ab65308). Deferoxamine (HY-B1625, purity 99.2%) and neocuproine (HY-W004563, purity 99.82%) were provided by MedChemExpress.

2) In lines 153-155, the authors stated: “Mn3O4 NPs did not trigger spontaneous hemolysis at concentrations of up to 40 mg/L (Figure 3A)”. Therefore, in the rest of experiments, authors used this concentration and higher to induce eryptosis. This is a huge concentration of NPs, and support that toxicity might be associated with other metals present in their composition. The authors need to characterize their nanoparticles with respect to other impurities (metals), for example using GC-MS, and prove that they are not in charge of this toxicity.

Answer: Thank you for pointing this out. We analyzed metal impurities in the Mn₃O₄ NP solutions using optical emission spectroscopy with inductively coupled plasma (ICP-OES). The results are presented in the Supplementary file as Table S1 and indicate that the concentrations of other metals in the solutions were negligible and could not be responsible for the observed effects.

3) To confirm that observed toxicity, oxidative stress and cell death parameters are induced by Mn3O4 NPs and not associated with other metal contaminants, the authors should perform experiments using chelators that are probable in charge of this effect (such as iron or cupper). For this purpose, authors are suggested to use deferoxamine or neocuproine or other specific metal/chelators to avoid Fenton/Haber–Weiss reactions that are probably responsible for the NPs toxicity.

Answer: Thank you for pointing this out. We performed extra experiments with the both suggested chelators, which did not affect the degree of Mn₃O₄ NPs-induced eryptosis. Data are presented in Figure S1.

4) It is highly recommended to characterize the pathways involved in the oxidative stress induced by NPs. This characterization require identification and use of electron paramagnetic resonance to detect specific fingerprints to dissect the radical pathway in charge of the oxidative stress.

Answer: Thank you for pointing this out. We agree that these data might provide extra insights into the erythrotoxicity of Mn₃O₄ NPs. However, we believe that the main aim of this study is to demonstrate that their toxicity might be associated with induction of eryptosis. Elucidation of ROS-specific pathways of Mn₃O₄ NPs-induced eryptosis is a challenging task that exceeds the aims of this study, since in addition to direct ROS generation promoted by nanomaterials per se, ROS accumulation in erythrocytes might be secondary and attributed to hemoglobin autooxidation, activation of NADPH oxidase or xanthine oxidase. Likewise, manganese ions release from NPs can contribute to ROS production. Therefore, in this manuscript, we deliberately tried to avoid delving into the mechanisms involved in oxidative stress that mediates Mn₃O₄ NPs-induced eryptosis.

We would like to express our sincere gratitude to the Reviewer for informative comments that have helped improve the quality of the manuscript. The corrections are made in the Track Changes mode.

Round 2

Reviewer 2 Report

Comments and Suggestions for Authors

Thank you for your response to the reviewers' questions. The reviewer appreciates the authors' efforts in refining the manuscript to address the concerns.

I suggest accepting this manuscript in the present form.